# Food for thought: The impact of short term fasting on cognitive ability

**Austin Landini**[1,2]*, **Michelle Segovia**[3], **Marco Palma**[4], **Rodolfo M. Nayga Jr.**[4,5]

**1** Division of Applied Social Science, University of Missouri, Columbia, MO, United States of America,
**2** Department of Economics, University of Missouri, Columbia, MO, United States of America, **3** Department of Economics, University of Delaware, Newark, DE, United States of America, **4** Department of Agricultural Economics, Texas A&M University, College Station, TX, United States of America, **5** Department of Food and Resource Economics, Korea University, Seoul, South Korea

* Austin.Landini@missouri.edu

## Abstract

Growing evidence suggests that resource scarcity can severely impede individuals' cognitive capacity, resulting in sub-optimal decision making. Few experimental studies investigate whether *food deprivation* as a form of resource *scarcity* influences decisions in other non-hunger related domains. We examine the effect of short term fasting on cognitive capacity by exogenously manipulating individuals' fasting time in a laboratory experiment. Participants were randomly assigned to one of three treatments: 1) 3-hour fast; 2) 12-hour fast; and 3) control, in which participants were not required to fast and consumed a protein shake upon arriving to the lab. Following the manipulation, participants completed the Raven's Progressive Matrices test which measures cognitive function. Although we find null treatment effects on cognitive ability, our results provide evidence that short term fasting does not directly inhibit cognition.

## Introduction

Recent additions to behavioral literature link resource scarcity to declines in both cognitive performance and productivity [1–3]. Food deprivation is an important form of resource scarcity. In this study, we examine the impact of a temporary fasting state on individuals' cognitive ability. Specifically, we test whether cognitive capacity is impaired due hunger induced by short term fasting, and whether the effects vary across individuals by characteristics such as Body Mass Index (BMI), race and their implicit association between healthiness and tastiness of food.

In understanding how temporary fasting impacts cognitive ability, we consider two prominent but contrasting hypotheses on how fasting impacts behavior. First, it is possible that fasting forces the brain to focus its limited cognitive resources on managing the state of food deprivation [3–5] resulting in a measurable decline in cognitive ability on unrelated tasks. In contrast to longer term fasting or food deprivation which may alter metabolistic functions, short term fasting is more likely to induce a hot state of hunger, which may have the potential to distract from careful decision making or calculation. Alternatively, there is some evidence that, hungry individuals actually make more careful choices [6], especially regarding food [7].

**Data Availability Statement:** Data in final processed form are available in a public GitHub repository at: https://github.com/azl0022/Food-for-Thought.git.

**Funding:** The author(s) received no specific funding for this work.

**Competing interests:** The authors have declared that no competing interests exist.

To examine the relationship between short term fasting fasting and cognitive ability, we exogenously manipulate individuals' fasting times in a laboratory experiment with 245 participants at a large university in the US South region. Participants were randomly assigned to one of three treatments: 1) 3-hour fasting treatment; 2) 12-hour fasting treatment; and 3) control, in which participants were not required to fast, and consumed a protein shake at the beginning of the session. Following the manipulation, all participants completed the Raven's progressive matrices test which measures fluid intelligence independent of acquired knowledge, followed by an implicit association test and a demographic survey which provided control variables for participants in the study.

This study contributes to a growing academic debate on whether there are cognitive penalties associated with resource scarcity. We examine the extent to which short term fasting impacts cognitive ability measured in a controlled setting. While previous studies have found that going periods without meals may negatively impact non-hunger related decisions such as leniency of court rulings [8], and monetary impatience [9], we find no significant effect of short run fasting on cognitive ability.

Our results provide insights on the (lack of) cognitive implications for short term fasting. In finding a null impact, we also contribute to a larger discussion of the role of scarcity on decision making. We provide evidence that there is a timing aspect over which food deprivation impacts cognitive ability. There has been extensive research linking long term food deprivation to reductions in cognitive ability [10–12]. Likewise, shorter term fasts may diminish individuals ability to sustain cognitive tasks over longer time periods [13, 14]. In contrast to studies on persistent food scarcity, and on longer term tasks following fasting, we do not find evidence that fasting for a few hours impacts performance on short run cognitive tasks. This finding is an important addition to the literature on short term, restricted hour intermittent fasting, which has been associated with potential health benefits such as weight loss and cholesterol reduction [15, 16], but is limited in its discussion of cognitive benefits or penalties.

Individuals may experience a fasting state for both involuntary and voluntary reasons. Involuntary reasons include food insecurity which may lead to malnutrition [17, 18]. This type of meal skpping is often chronic and longer term. Short term fasts are often voluntary and reasons for voluntary fasting include religious practices and health and dietary needs. Short term fasting is common during certain religious periods. For example, there are about 1.9 billion followers of Islamic religions worldwide, with the majority fasting during Ramadan. Fasting is also common amongst Christians, including Greek Orthodox Christians who may fast for up to 180–200 days per year due to a variety of observances such as lent [15]. Individuals may also fast as a means of dieting. Johns Hopkins Medicine defines intermittent fasting as an eating plan that switches between fasting and eating on a regular schedule. Forms may include full day fasts or restrictions on hours of calorie intake. A 2010 survey conducted by the Booth Research Services notes that about one in three US adults were attempting to lose weight by means of dieting. Among those, about 17% chose to diet by practicing intermittent fasting methods.

Previous health and medical literature has detailed the potential benefits associated with an intermittent fasting diet. Despite its popularity, much of the scientific evidence for the health benefits of intermittent fasting is extrapolated from animal studies. Patterson, et al. [19] identified two human intervention studies that tested for the health impacts of a fasting regimen which limited eating at night before the study. Results from both studies found significant reductions in weight following the fast. Likewise, a crossover study tested biomarkers related to fasting and found reductions in fasting glucose and improvements in cholesterol counts [20–22].

We distinguish our results from previous studies examining the impact of long term malnutrition on cognitive ability. Previous research has shown that long-term malnutrition can negatively impact short-term performance on cognitive evaluations and in the classroom [10] and on cognitive tests such as the Raven's Progressive Matrices [11]. Some of these findings are confirmed in studies examining shorter term fasting. For example, skipping breakfast has been shown to induce cognitive penalties upon school children [13, 14]. However, throughout the literature, there is no clear consensus on the impact of short term fasting on cognitive ability (Pollitt, 1998).

Benau, et al. [23] reviewed ten studies on the effects of intermittent fasting on cognition in healthy adults. These studies do not reach a consensus, with some finding no cognitive penalty associated with intermittent fasting, and others finding negative effects on executive function, psychomotor speed, and mental rotation. In response, Gudden, et al. [24] concluded that so far, no convincing direct effects of intermittent fasting on cognition in healthy adults has been found. Even where cognitive impairments resulting from short term fasts have been detected [25, 26], the effects tend to be small. In a follow up study, Green, et al. [27] randomized their respondents into four different levels of fasting lengths. Respondents completed cognitive ability tasks including a Bakan vigilance task, reaction time test, focused attention task, tapping task, and recall task. Across the cogntive ability tests, the authors found no significant effects of fasting up to 24 hours, other than that the longest fasting group had the slowest response rate in the tapping test. This work indicates that, contrary to what is often assumed, short-term food deprivation per se does not substantially affect cognitive function [27–29]. Still, there is limited academic evidence concerning whether hunger induced by fasting impacts short-term cognitive ability. Given the limited and inconclusive results of studies on the relationship between intermittent fasting and cogntitive ability, Cherif, et al. [30] concluded that "several studies have demonstrated that long-term food restriction was associated with impairments in cognitive function, including poor performance on a sustained attention task. However, other studies have shown that memory performance significantly improved during fasting. In the investigation of mechanisms by which dietary restriction acts on cognitive function and to determine how these diets work, further detailed and unified research studies are necessary" (Cherif, et al., pg. 44).

Given the potential health benefits associated with short term fasting, recent attention has been given to its impact on behaviors in contexts other than food and health. For example, previous experimental studies have suggested that hunger induced by fasting can negatively affect individual decision making. In a lab experiment, Ashton [9] found that participants who fasted for three hours had a significant preference for choices that involved an immediate reward. Kuhn, et al. [31] (2014) found a similar treatment effect among glucose deprived subjects, while Levy et al. [32] found that hunger is associated with greater financial risk tolerance. Danziger, et al. [8] suggested that judicial rulings can be swayed by extraneous variables that should have no bearing on legal decisions, such as breaks to eat. The authors found that in sessions before meals, the percentage of favorable rulings gradually declines from 65% to near 0%, then abruptly returns to 65% after a meal break. Given this evidence, we explore the possibility that short term fasting impairs decision making via mental depletion, leaving fewer resources for other tasks, but do not find evidence of this effect when participants completed short cognitive tasks.

Conversely, De Ridder, et al. [6] examined the counter-intuitive hypothesis that a state of hunger may improve decision making. Their study, composed of three lab experiments, finds that greater appetite leads to more advantageous choices. Hungry individuals performed better on a complex task taken from the Iowa Gambling Task assessment. Additionally, hungry participants chose larger future rewards in a delayed discounting game and were less likely to take

gambles than satiated individuals. Likewise, Carvalho, et al. [33] found no effect of financial constraint, which is often associated with meal skipping, on financial risk taking and cognitive ability. Radel and Clement-Guillotin [7] found that when deprived of food, participants perform better on food related cognitive tasks, but that this benefit does not extend to unrelated tasks. One possible explanation is that during a state of hunger, individuals tunnel their mental resources on the scarcity, leading to improved decision making on food related tasks. It is also possible that sated individuals may be lethargic relative to those experiencing a short term hunger state, and therefore make less careful decisions [6]. Our results suggest that any such penalties do not apply in the case of short cognitive exercises completed after a short fasting period of up to about 12 hours.

Finally, there are only a few studies testing for heterogeneous effects of fasting on cognitive ability across individuals with different characteristics. Chen, et al. [34] examined whether effects of an 8–12 hour fast on economic decision making vary by gender using a lab-in-the-field experiment. The authors compared hungry and sated respondents across a variety of measures including cognitive performance, as measured by a memory reflection test highly correlated with standardized test performance. Their study found that men and women respond differently to the fasting treatment. When the questions on the cognitive test were non-food framed, men consistently scored higher than women, but this gender gap disappeared when the questions were food-framed. This result suggests that there is a potential for heterogeneous treatment effects from fasting across different individuals. Similarly, Segovia, Palma, and Nayga [35] investigate the effect of food anticipation on cognitive function across individuals with varying weight status. The authors find that an anticipatory food effect can help offset the cognitive cost associated with hunger by enhancing the mental resources of overweight and obese individuals, but do not find this effect among normal weight participants. While these previous studies suggest that there may be an individual varying connection between food deprivation induced via short term fasting and decision making, we find only limited evidence of heterogeneous treatment effects in our study.

## Materials and methods

### Participants

Two hundred forty-five students and university employees participated in this experiment in exchange for $10 cash compensation. The experiment was approved by the University's Institutional Review Board and data was collected from February 23 to May 22 2018. All participants signed an informed consent statement.

To ensure balance in the number of normal weight, overweight, and obese individuals by treatment, a pre-screening survey was used during recruitment. Individuals with a history of eating disorder, diabetes, or any dietary restrictions did not qualify as participants. The pre-screening survey was sent by bulk email to all students and staff members on campus and it contained socio-demographic questions including weight and height measures. Although self-reported measures of height and weight were collected in the pre-screening survey, these measures were also recorded by experimenters at the end of the session for accurate classification of weight status. To increase enrollment, in-person recruiting was also conducted across campus.

To ensure that our study had sufficient power to detect meaningful differences in the number of correct responses on the Raven's Progressive Matrices test, we conducted a power analysis based on a Poisson distribution with confidence level $\alpha = 0.05$. Using the control and treated means as the null and alternative hypotheses with a sample size N = 244, the results indicated that the study had an estimated power of $1 - \beta = 99.57\%$ to detect this difference.

Thus, the study is well-powered to identify differences in cognitive performance as measured by the Raven's test.

## Ethics statement

Approval for an experiment with human subjects was granted by the Texas A&M University Institutional Review Board (protocol #IRB2017–0896D). Data were analyzed anonymously.

## Experimental design

We manipulated the number of fasting hours prior to the experiment, which generated the control and treatment groups needed to estimate the effect of short term fasting on cognitive ability. One out of 245 respondents did not fully complete the survey, thus the observation was dropped from analysis. Participants were randomly assigned to one of three conditions: 1) *3-hour fasting* condition (N = 91), 2) *12-hour fasting* condition (N = 79), and 3) Control condition (N = 75). The 12-hour fasting time chosen reflects recent research suggesting that at about 12 hours of fasting there is a metabolic shift from lipid/cholesterol synthesis and fat storage to mobilization of fat through fatty acid oxidization and fatty-acid derived keytones. The 12-hour fast reflects recent research suggesting that at about 12 hours of fasting there is a metabolic shift from lipid/cholesterol synthesis and fat storage to mobilization of fat through fatty acid oxidization and fatty-acid derived keytones [36]. This metabolic shift could induce fatigue or other factors which might impact performance. Even before any metabolic change has time to take effect, it is possible that hunger may affect decision making or cognitive performance because it is associated with activation of brain areas that are disproportionately activated when immediate rewards are available [9].

Participants in the 3-hour fasting condition were asked to refrain from eating for at least three hours prior to their session with the goal of mimicking the state of hunger that individuals often experience in between meals, similar to the judges studied in Danziger, et al. [8]. On average, participants in this group reported fasting for 8.36 hours (s.d. = 3.84 hours). Participants in the 12-hour fasting condition were instructed to refrain from eating past midnight before the experiment. Participants in this group reported fasting for 10.18 hours on average (s.d. = 3.61 hours). In the control condition, fasting was not required and participants were given a protein shake that had to be consumed before the experimental tasks began. The goal of the protein shake was to manipulate the satiation level of the control group, using a highly-satiating nutrient [9]. The 12 ounce Special K brand breakfast shake contained 30g Protein, 190 Calories, 5g Total Fat, 24g Total Carbohydrate and 18g Total Sugars.

As manipulation checks, our participants self-reported number of fasting hours and level of hunger in the post experiment survey. While the fasting treatments failed to create a significant difference in fasting hours or hunger between the two levels of fasting, the experiment did succeed in manipulating the hunger levels of participants as compared to the control group (difference significant at the $p < .01$ significance level), and participants in the two treated groups fasted for 9.2 hours (s.e. 3.83 hours). Following the suggestions of both referees, we have combined the treatment groups in our main analysis.

## Procedures

The experiment took place in a computer lab at a large public university in the South region of the US. Most experimental sessions took place in the morning. The median session size was seven participants and all sessions were conducted by the same experimenter. Upon arrival to the lab, participants were provided with an ID number to be used throughout the session in order to maintain anonymity. Then, they were randomly assigned to a computer station where

they read and sign in an informed consent. Next, participants in the control were provided with a 10 ounce protein shake and were instructed to consume all of its content before the experimental tasks began. The protein shake was available in two flavors (vanilla and strawberry) and participants were free to choose based on their preference.

The instructions and tasks were administered to participants through Qualtrics; however, participants were given the opportunity to ask questions to session monitors at any stage of the experiment. To begin, all participants completed a cognitive performance test lasting 16 minutes. Afterward, participants completed an implicit association test and filled out a post-experiment survey, which collected information on demographics and behavioral characteristics used as control variables in the study. At the end of the experiment, two experimenters measured participants' weight and height. This was done outside the lab using a digital scale and a meter. Finally, participants filled out a receipt form and received their payment in cash. In summary, participants in all conditions performed three computer-based tasks in the following order: 1) cognitive performance test, 2) implicit association test, and 3) post-experiment survey. The implicit association test and post-experiment survey provide control variables about the research participants which are used in our analysis.

## Cognitive performance test

We administered twenty four problems from the Raven's Standard Progressive Matrices to measure participants' cognitive ability. The Raven's test is used to assess ability associated with abstract reasoning and is considered a nonverbal estimate of fluid intelligence [37]. The problems were displayed individually in order of difficulty. In each problem, participants were asked to analyze a geometric pattern and identify the missing element that completed the pattern of shapes. After observing an example, participants had 16 minutes to answer 24 problems. The standard Raven's Progressive Matrix test contains five sets of 12 problems each, and is usually administered over 40 minutes. Due to time constraints, we chose to administer two of the five sets from the standard test. The sets are the same as used in Segovia, et al. [35]. Once the 16 minutes elapsed, they automatically proceeded to the implicit association task. Selected problems were pilot tested among students to ensure increasing difficulty of the problems and understanding of instructions.

Previous economic studies have used Raven's Progressive Matrices to study whether anticipation of food intake can improve cognitive performance [35], assess the effect of working memory capacity on behavioral task performance under cognitive load [38] and in single price auctions [39] and to study price salience in food choices [40]. Fig 1 illustrates the first question from the Raven's Progressive Matrices test included in this task.

## IAT and post-experiment survey

The Implicit Association Test (IAT) is a measure widely used by psychologists to measure an individual's automatic association (or implicit cognition) between mental representations of objects or concepts in memory [41]. That is, the test measures memory associations of which an individual has no conscious awareness.

In this study, the IAT was designed to measure participants' implicit association between healthiness and tastiness of food products. The test was presented as a computerized categorization task with no time restriction to be completed. The IAT rests on the assumption that it should be easier to have the same behavioral response (i.e., a key press) to concepts that are strongly associated than to concepts that are weakly associated in the memory. In theory, an individual can more rapidly sort stimuli when pairings are compatible with one's implicit associations. This tool has been used to measure mental association between food healthiness and

Question 1.

What is the missing element?

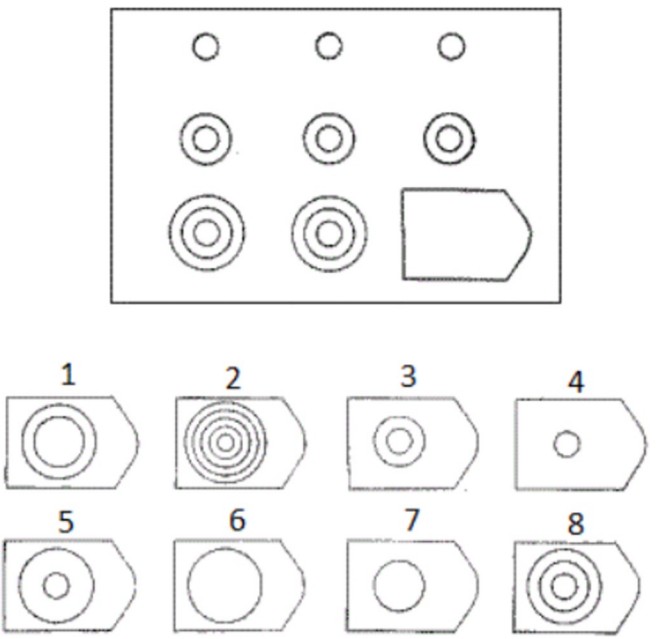

**Fig 1. Question 1 from the Raven's Progressive Matrices test.**

tastiness in a number of previous studies [42–44]. Recent literature has also used implicit association testing to study the association between tastiness and sustainability among meat and meatless alternatives [45]. It has been well established that explicit and implicit attitudes combine to have a significant role in determining human behavior [46–48]. However, research on how implicit and explicit processes combine to control overt responses is still in progress [49]. We hypothesized that implicit association could be a mediator of the relationship between short term fasting and Raven's score, but found no statistically significant relationship.

The post-experiment survey collected information regarding socio-demographic and behavioral characteristics of the participants. Since participants in the treatments were asked to fast for specific hours prior to their experimental session, we included questions that helped us identify individuals that did not comply with the fasting requirements. Specifically, we asked participants to report the time since their last meal, which we translated into a total fasting time variable, and rate their level of hunger using a 9- point rating scale (1 = "Not at all" and 9= "Extremely hungry") [9]. The post-experimental survey is the basis for collection of individual specific controls used in our analysis and relies on self reported measures except for BMI. Summary statistics of the survey responses are provided in Table 1. The primary purpose of collecting the sociodemographic covariates was to test for heterogeneous treatment effects following Chen, et al. [34] and Segovia, et al. [35].

## Results

### Sample description

Out of 245 participants, 244 completed all three experimental tasks, resulting in 74, 91, and 79 participants in the *control*, *3-hr fasting*, and *12-hr fasting* treatments, respectively. When

**Table 1. Summary statistics.**

| VARIABLES | (1) Control mean (sd) | (2) 3-hr Fast mean (sd) | (3) 12-hr Fast mean (sd) | (4) Treated mean (sd) | (a) Difference (4)-(1) (p-value) | (b) p-value (Kruskal-Wallis) | (c) p-value (Mann-Whitney) |
|---|---|---|---|---|---|---|---|
| BMI | 25.98 | 25.60 | 25.35 | 25.51 | -.466 | .561 | .284 |
|  | (4.950) | (5.307) | (4.634) | (4.990) | (.501) |  |  |
| Male | 0.311 | 0.411 | 0.278 | 0.353 | .042 | .257 | .524 |
|  | (0.466) | (0.495) | (0.451) | (0.478) | (.525) |  |  |
| Exercise | 2.426 | 2.618 | 2.288 | 2.465 | .039 | .902 | .968 |
|  | (1.814) | (2.073) | (1.846) | (1.972) | (.884) |  |  |
| Sleep | 6.799 | 6.322 | 6.430 | 6.382 | -.416 | .301 | .161 |
|  | (1.089) | (1.636) | (1.896) | (1.757) | (.060) |  |  |
| Breakfast | 4.689 | 4.600 | 4.051 | 4.318 | -.372 | .176 | .272 |
|  | (2.220) | (2.326) | (2.325) | (2.335) | (.250) |  |  |
| Fasting Hrs. | 0.000 | 8.363 | 10.177 | 9.206 |  |  |  |
|  | (0.000) | (3.841) | (3.611) | (3.834) |  |  |  |
| Hungry | 3.149 | 5.055 | 5.139 | 5.094 | 1.945 | .001 | .001 |
|  | (2.345) | (2.238) | (2.135) | (2.184) | (.001) |  |  |
| Income (thousands) | 67.57 | 61.11 | 68.54 | 64.59 | -2.982 | .858 | .866 |
|  | (54.90) | (52.06) | (47.64) | (50.03) | (.679) |  |  |
| Age | 25.42 | 23.19 | 21.46 | 22.38 | -3.034 | .235 | .229 |
|  | (12.69) | (8.43) | (5.59) | (7.38) | (.019) |  |  |
| White | 0.581 | 0.473 | 0.456 | 0.465 | -.116 | .332 | .088 |
|  | (0.497) | (0.502) | (0.501) | (0.500) | (.095) |  |  |
| Black | 0.081 | 0.0440 | 0.152 | 0.094 | .012 | .468 | .754 |
|  | (0.275) | (0.206) | (0.361) | (0.292) | (.755) |  |  |
| Hispanic | 0.189 | 0.253 | 0.253 | 0.251 | .062 | .741 | .291 |
|  | (0.394) | (0.437) | (0.438) | (0.435) | (.291) |  |  |
| Asian | 0.149 | 0.185 | 0.114 | 0.152 | 0.003 | .966 | .956 |
|  | (0.358) | (0.390) | (0.320) | (0.360) | (.946) |  |  |
| All Other | 0.000 | 0.044 | 0.025 | 0.035 |  |  |  |
|  | (0.000) | (0.206) | (0.158) | (0.185) |  |  |  |
| NonWhite | 0.419 | 0.532 | 0.544 | 0.538 | .119 |  |  |
|  | (0.497) | (0.502) | (0.501) | (0.500) | (.088) |  |  |
| N = | 74 | 91 | 79 | 170 |  |  |  |

looking at compliance with the fasting requirement, we find that individuals in the *3-hr* and *12-hr* fasting treatments did not differ significantly in their reported fasting times. The mean fasting time period in the *3-hr* fasting treatment was 8.3 hours (s.d. = 3.84 hours), which is close to the average fasting time of 10.2 hours (s.d. = 3.61 hours) reported in the *12-hr* fasting treatment. S1 Fig in S1 Appendix displays histograms with the frequency of reported fasting hours by treatment.

Table 1 reports p-values between treatments from mean comparisons of demographic variables using Kruskal-Wallis tests for the individual treatments and Mann-Whitney tests for the combined treatment groups. Results provide evidence of a balanced sample ($p \geq 0.10$ for all tests), suggesting proper participant randomization into treatments. All covariates from the survey are self-reported except for height and weight. Among the participants, 66% were female, with an average age of 23 years old. About 77.5% of participants were undergraduate

students (21 or younger), with the remainder of the sample composed of graduate students, faculty and staff members, ranging in ages 22–80 years old. Approximately 50% of respondents identified themselves as White, 23% as Hispanic, 15% as Asian, 8% as Black, and 4% as Native or Other race. Research assistants measured participants' height and weight, which is used to calculate BMI. All other measures are self-reported in the post experiment survey.

Respondents indicated that they ate breakfast an average of 4.69 days per week (s.d. = 2.22 days) in the post experiment survey, which is consistent with recent literature on meal skipping tendencies among youth and young adults [50]. Respondents indicated that they exercise approximately 2.43 days per week (s.d. = 1.81 days) and slept for an average of 6.80 hours (s.d. = 1.09 hours) the night before the experiment. The control group who had consumed the protein shake, indicated an average hunger level of 3.15 (s.d. = 2.35) on a 1–9 scale, while the treatment groups indicated an average hunger level of 5.09 (s.d. = 2.18); with the 3 and 12-hour fasting groups self reporting approximately equivalent hunger levels of 5.06 (s.d. = 2.24) and 5.14 (s.d. = 2.14), respectively. This indicates that both treatments were generally successful at exogenously manipulating participants' hunger levels via short term fasting.

## Data analysis

We test whether short-term induced hunger as a result of fasting (treatment) has an impact on cognitive ability as measured by performance on a Raven's test. First, we calculate a test score by dividing the subject's number of correct answers by a total 24 possible questions. Questions which were not answered are counted as incorrect responses. To prepare our data for analysis, we combined the results of the Raven's test with IAT test D-Score, recorded height and weight, and self-reported sociodemographic survey results. We calculate BMI using the standard formula: $\frac{weight}{height^2} * 703$. The complete merged data including all self-reported covariates is available with the experimental protocol and a readme file detailing the variables used for analysis on a public GitHub repository. No identifying information about respondents is included in the final dataset.

While the Raven's score is calculated as a percent, it is not a continuous variable, rather it reflects counts of correct answers. As a result, the errors are not normally distributed and a linear model is not the best fit. Instead we estimated a Quasi-Poisson regression model with a log link function to model the expected number of correct answers on the Raven's Progressive Matrices test in STATA. The log link function takes into account the fact that there are no negative scores and percentage correct is bounded between [0, 1]. The model was specified as:

$$log(\mathbb{E}[\text{Raven's Score}_i]) = \beta_0 + \beta_k \text{Treatment}_i + \beta_n \chi_i \qquad (1)$$

where coefficients $\beta_k$ test for the impact of the two fasting treatments, and $\beta_n$ measure the effect of other control variables: income, exercise, previous night's sleep hours, gender, age, race and recorded BMI. The individual specific behavioral and socio-demographic characteristics, $\chi_i$, are described in Table 1. Because individuals assigned to the 3 and 12-hour fasting treatments did not vary significantly in their actual reported fasting time before the experiment, we combine the treatments in our main analysis. For robustness, we also provide specifications with the disaggregated treatments, and using total reported fasting hours in S2 Table in S1 Appendix.

## Treatment effects on Raven's test performance

Respondents were allotted 16 minutes to complete two blocks of 12 questions taken from the Raven's matrices tests. Previous literature has not reached a consensus on whether short term fasting imposes a cognitive penalty on individuals. Mullainathan [5] found that "[. . .] we can

directly measure mental capacity or, as we call it, bandwidth. We can measure fluid intelligence, a key resource that affects how we process information and make decisions. We can measure executive control, a key resource that affects how impulsively we behave. And we find that scarcity reduces all these components of bandwidth— it makes us less insightful, less forward- thinking, less controlled" (Mullainathan 2013, pg. 13).

While short term fasting may reduce performance on non-food related tasks [8, 9], there is some contrasting evidence which has found short term cognitive benefits of fasting, as compared to sated individuals [6, 34]. We hypothesized that fasting participants would have a more difficult time focusing on the cognitive ability task, resulting in lower average scores. However, we do not find a statistically significant difference on short run cognitive ability in the treated group. That is we find no statistically significant evidence of a relationship between short term fasting and cognitive ability. Since the two treatment groups did not significantly differ in their average total fasting times, for our main analysis in Table 2, we combine the two treatments. S2 Table in S1 Appendix presents the results by treatment. As a robustness check, we also include specifications with total fasting hours rather than treatment.

**Table 2. Quasi-Poisson regressions on Raven's test score.**

|  | (1) Raven's Score | (2) Raven's Score |
|---|---|---|
| Treated | 0.033 | 0.035 |
|  | (0.028) | (0.024) |
| IAT D-Score |  | -0.028 |
|  |  | (0.018) |
| Income |  | -0.001 |
|  |  | (0.000) |
| Exercise |  | 0.005 |
|  |  | (0.006) |
| Sleep |  | 0.007 |
|  |  | (0.007) |
| BMI |  | 0.001 |
|  |  | (0.002) |
| Male |  | 0.068*** |
|  |  | (0.022) |
| Age |  | -0.008*** |
|  |  | (0.002) |
| White |  | 0.132*** |
|  |  | (0.046) |
| Asian |  | 0.138*** |
|  |  | (0.048) |
| Hispanic |  | 0.020 |
|  |  | (0.046) |
| Constant | -0.336*** | -0.336*** |
|  | (0.024) | (0.097) |
| N | 245 | 240 |
| Residual Deviance | 4.625 | 6.625 |

Robust Standard errors in parentheses

$*p < .1$,

$**p < .05$,

$***p < .01$

Previous literature has suggested that it may be possible for fasting to impose a bandwidth tax upon the cognitive functioning of the decision maker by reducing focus and attention available for decisions and thoughts outside of managing scarcity. However, in contrast to studies which find a significant negative relationship between long term hunger and cognitive ability, or school performance measured throughout the day, our study provides no evidence of such effects associated with short fasting periods. Nonetheless, our study provides important empirical evidence regarding the short term impact of fasting on cognitive ability. While previous literature suggests that skipping meals may impede learning activity throughout the day [10, 11, 13, 14], it does not appear to inhibit cognitive function in a short term task. This is likely due to the many confounding factors in the different practices associated with short term or intermittent fasting.

Table 2 shows results for Poisson regressions on the Raven's test score with and without controls. We do not find a significant effect of the fasting treatment on short term cognitive ability. While there is a weakly significant positive coefficient on the impact of a short term fast on Raven's test score, this effect is not significant when controlling for individual specific characteristics. Participation in the treated group is associated with an score increase of 3.5% on the Raven's test (s.e. 2.4%, $p \approx .13$). Similarly, in S2 Table in S1 Appendix we find that neither the disaggregated treatments nor total fasting hours is significantly associated with Raven's score.

With regard to BMI, there is growing evidence that obesity is linked to adverse neurocognitive outcomes including reduced cognitive function [51]. A study of healthy working age adults found that in a cross-section, a higher BMI was associated with lower cognitive scores after adjusting for age, sex, educational level, blood pressure, diabetes, and other psycho-social characteristics [52]. We test for the relationship between weight and cognitive ability across the entire BMI distribution in Table 2, but do not find a significant effect. The coefficient on BMI on the model is not statistically distinguishable from zero ($p \approx .94$).

We do find statistically significant effects of socio-demographics on cognitive ability. Previous research has not reached a consensus on gender differences in the Raven's Progressive Matrices test. While previous studies have found evidence that men and women use different reasoning processes and brain functions to solve the problems, their performance is relatively similar [53]. Mackintosh and Bennett [54] found that men outperformed women on some types of reasoning skills required to determine the answers, while Savage-Mcglynn [55] found that age matters, with girls outperforming boys until age 15, and the reverse effects found afterwards. In our study, males performed about 7% better than female participants on the Raven's Progressive Matrices tests (s.e. 2.2%, $p < .01$).

Previous studies have also reported that young respondents tend to outperform older ones on the Ravens Progressive Matrices test [56, 57]. The results of our study are consistent with this finding. We find a small, but slightly significant negative effect of age on Raven's test score. Each year of age decreased expected score by 0.8% ((s.e. 0.2%, $p < .01$)). It is possible that older participants might be more sensitive to the fasting treatment. Since the majority of our sample was composed of college-aged individuals, we suggest that future studies should consider more carefully the possibility for heterogeneous treatment effects across age groups.

In addition, previous studies have found a performance gap on the Raven's Progressive Matrices test by race, especially between White and Black respondents [58]. However, it has also been argued that some of the racial gap in cognitive test performance is attributed to a 'stereotype threat', describing a decrement in test performance that results when members of some group fear that their test performance will confirm a negative stereotype of their group [59, 60]. We do find a significant racial gap in Raven's test performance, with White and Asian respondents' expected scores about 12–13% higher on average than Hispanic, and Black or

Other race respondents (s.e. 4.6% for White and 4.8% for Asian, p < .01 for both groups). We do not find a significant difference in performance between White and Asian participants.

## Limitations

While the Raven's progressive matrices test has been widely used in previous studies of cognitive ability, it is limited in its scope and may be a relatively narrow measure of cognitive ability. The test measures fluid intelligence, which is a non-verbal estimate of abstract reasoning, cognitive thinking, problem-solving, and pattern recognition skills. Previous studies have highlighted aspects of cognitive ability such as discount rate or impulsiveness which may be salient in overall cognition [6, 8, 9, 31, 32]. Other factors such as memory, attention, or executive function, could also be affected by fasting. This study is limited by the number of tasks we could reasonably ask our respondents to complete during the experimental session, however, future studies on the impact of fasting on cognitive ability should consider aspects of the relationship other than logic or fluid intelligence.

Our socio-demographic survey included respondent age. While most participants were college students, the sample also included general population participants. There may be some concern that college students are more accustomed to meal skipping throughout the day and therefore less sensitive to the fasting times in this study. We do find a negative relationship between the age of our respondents and their score on the Raven's test indicating that it is possible that the cognitive response of older respondents may be more severely impacted by fasting. It is also possible that cognitive ability may be correlated with age. Unfortunately, we do not have the power to test effects by age. Future studies should continue to study this type of heterogeneous treatment effect across respondents

While randomization into the two treatments created an exogenous variation (p<0.01) in hunger level between the treated (mean = 3.15, s.e. = 2.35) and control groups (mean = 5.09, s.e. = 2.18), there was not a significant difference in the fasting time between the two treatment groups. This is likely a result of the experimental sessions being conducted in the morning. As a result, we are unable to comment on whether there are significant differences between those who skip a single meal and those who engage in slightly longer fasts which might create some metabolic change.

## Conclusions

Despite the popularity of intermittent fasting for religious purposes, and more recently as a dietary alternative, there are few studies which investigate the link between fasting and cognitive ability. Previous studies describe the propensity for fasting to alter short term decision making [6, 8, 9, 31, 32]. The short term decisions described in these studies, such as monetary patience and judicial rulings, are both tasks which require sustained attention. For example, to make well informed financial decisions, one needs to research alternatives or consult with an advisor. Likewise, the judges studied in Danziger, et al. [8] required longer term attention span to internalize and process facts presented in cases. As we find no effects for the impact of short-run fasting on short- run cognitive function, our results are important because they provide evidence that fasting should only inhibit cognitive function when either the fasting or cognitive tasks occur over extended periods of time.

Sustained long-run food deprivation (malnutrition) reduces performance on tests measuring attention, working memory, learning and memory and visuospatial ability [12]. These long run impacts of undernourishment could help to explain results such as reduced performance in Kenyan children on Raven's matrices exams [11]. However, there has been little experimental research evaluating the impact of intentional short term fasting, common

for religious and dietary purposes, on cognitive ability and its implications on decision making.

We exogenously manipulate individuals' fasting times in a laboratory experiment to test the impact of fasting on cognitive ability using the Raven's Progressive Matrices test. While we do not find significant effects associated with the fasting treatment, the study provides additional evidence that individuals who choose to infrequently skip meals for religious, dietary, or other voluntary purposes should not face a significant short run cognitive penalty associated with this behavior. We test for heterogeneous treatment effects across important socio-demographic features such as BMI, income, race and implicit association. We observe the entire distribution of respondents' BMI status, rather than categorizing into obese/non-obese, but do not find a significant treatment effect. We do find that White and Asian respondents scored slightly higher on the Raven's test than Hispanic and Other race respondents, and that male respondents outperformed female. We also find that age is negatively related to Raven's score.

While previous research has suggested that scarcity in general may impede decision making [4, 5] and that hunger, specifically, is a form of scarcity which may inhibit cognitive capacity [8, 9, 31, 32], we do not find significant cognitive effects related to a short term (8–12 hour) fasting treatment. Recent research also suggests that fasting may have a disparate impact on individuals with different characteristics [34], however, we do not find significant heterogenous treatment effects across respondents by BMI, race, or their implicit association between healthy and tasty food. On the contrary, there may be short run health and cognitive benefits [6, 15, 35]. While we are not able to verify any short run benefits of the fasting period, we conclude that short run fasting does not significantly inhibit short run cognitive ability.

## Supporting information

**S1 Appendix.**
(PDF)

## Acknowledgments

The authors would like to thank the editor and two anonymous referees at PLOS ONE for their invaluable feedback. We would also like to thank participants in the 2018 and 2023 SAEA annual meetings and 2018 TExAS conference for their discussion; and research assistants at the Human Behavior Lab for helping with data collection.

## Author Contributions

**Conceptualization:** Michelle Segovia, Marco Palma, Rodolfo M. Nayga, Jr.

**Data curation:** Michelle Segovia, Marco Palma, Rodolfo M. Nayga, Jr.

**Formal analysis:** Austin Landini.

**Investigation:** Austin Landini, Michelle Segovia, Marco Palma, Rodolfo M. Nayga, Jr.

**Methodology:** Michelle Segovia, Marco Palma, Rodolfo M. Nayga, Jr.

**Project administration:** Michelle Segovia, Marco Palma, Rodolfo M. Nayga, Jr.

**Resources:** Michelle Segovia, Marco Palma, Rodolfo M. Nayga, Jr.

**Supervision:** Michelle Segovia, Marco Palma, Rodolfo M. Nayga, Jr.

**Visualization:** Austin Landini.

**Writing – original draft:** Austin Landini.

**Writing – review & editing:** Michelle Segovia, Marco Palma, Rodolfo M. Nayga, Jr.

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
