## [Decision Letter · Decision Letter 0]

4 Sep 2024

PONE-D-24-28077Food for Thought: The Impact of Short Term Fasting on Cognitive AbilityPLOS ONE

Dear Dr. Landini,

Thank you for submitting your manuscript to PLOS ONE. After careful consideration, we feel that it has merit but does not fully meet PLOS ONE’s publication criteria as it currently stands. Therefore, we invite you to submit a revised version of the manuscript that addresses the points raised during the review process. I have now heard back from two knowledgeable referees who have published well in the area of intermittent fasting. The news is mixed. One referee recommends a "reject," while the other asks for a revision. While I am not an expert in the area, I could follow the concerns of the referees, esp. the more negative one. I am offering you a chance to revise the manuscript paying careful attention to the latter report. I realize some of the comments relate to "unfixable" matters -- the experiment has been done and one cannot go back and run it all over again. Keeping that in mind, I'd like you to defend your design choices carefully in a "discussion" section as well as in a "limitations" section. Both referees complain about the way the paper is structured or the way the results are presented; please fix that. I will not send the paper back to the referees, so you have one final chance to make your case to me for why the paper should be published. I like the paper and your design and do see a path to eventual publication provided you can convince me that you've paid attention to both referees' concerns and adequately defended the paper.

We look forward to receiving your revised manuscript.

Kind regards,

Joydeep Bhattacharya

Academic Editor

PLOS ONE

4. We note that you have referenced (Heather Schofield. The economic costs of low caloric intake: Evidence from india. Unpublished Manuscript, 2014) which has currently not yet been accepted for publication. Please remove this from your References and amend this to state in the body of your manuscript: (ie “Bewick et al. [Unpublished]”) as detailed online in our guide for authors

http://journals.plos.org/plosone/s/submission-guidelines#loc-reference-style.

Reviewers' comments:

Reviewer's Responses to Questions

**Comments to the Author**

1. Is the manuscript technically sound, and do the data support the conclusions?

Reviewer #1: No

Reviewer #2: Yes

2. Has the statistical analysis been performed appropriately and rigorously? 

Reviewer #1: No

Reviewer #2: Yes

3. Have the authors made all data underlying the findings in their manuscript fully available?

Reviewer #1: No

Reviewer #2: No

4. Is the manuscript presented in an intelligible fashion and written in standard English?

Reviewer #1: Yes

Reviewer #2: Yes

5. Review Comments to the Author

Reviewer #1: Thank you for an opportunity to review the manuscript titled: “Food for Thought: The Impact of Short-Term Fasting on Cognitive Ability.” The current study aims to investigate the effect of fasting on cognitive ability by manipulating fasting durations in a laboratory setting and measuring participants' performance on cognitive tasks. Overall, the study found no significant difference in cognitive ability across different fasting durations (3 hours, 12 hours, and control with a protein shake). Despite expectations, short-term fasting did not impact performance on the Raven's Progressive Matrices test.

The current manuscript has several strengths such as relatively large sample size and a robust literature review. However, there are also numerous weaknesses that I highlight below:

1. Manipulation failure. There was significant overlap in the fasting durations reported by participants in the short and long fasting groups, which may have diluted the differences between treatments. This overlap could be a result of participants' non-compliance with fasting instructions or variability in individual metabolism and meal timing. As such, the manipulation has failed. Relatedly, I did not see a solid argumentation behind why these specific desired fasting durations (3 vs 12 hours) were selected. Is there anything specific that happens in human physiology after having fasted for 3 hours relative to 12 hours? Even more importantly, is abstaining from eating for 3 hours indeed a “fasting”?

2. Limited scope of cognitive tasks. The study focused solely on the Raven's Progressive Matrices test, which is a relatively narrow measure of the cognitive ability. A stronger justification behind the sole focus on this specific task is needed to build an argument that fasting might (or might not) affect cognitive capabilities. What about other aspects of cognitive ability such as, for instance, discount rate or impulsiveness? In sum, This narrow focus might not capture other aspects of cognitive function, such as memory, attention, or executive function, which could also be affected by fasting.

3. Unclear role of IAT. This is related to point above. When reading the manuscript, I thought that the implicit association test (IAT) will be yet another outcome variable, potentially broadening the scope of the investigation. However, the results section does not focus on IAT at all.

4. Claims not supported by the findings. For example, the authors claim in the abstract that their results contribute to the literature on the health implications of short-term fasting, despite finding null treatment effects on cognitive ability. It is unclear how the current findings contribute to this research area, especially considering the fasting manipulation did not produce significant results. I found many other unsupported claims in different parts of the manuscript.

5. The lack of the overarching research goal. The manuscript lacks a clear research goal and fails to support several claims. The role of the Implicit Association Test (IAT) is unjustified, and the operationalization of cognitive ability is limited. The introduction does not clearly position the research within a specific field of study. While resource scarcity is mentioned, the predictor variable—fasting differing in duration (albeit being short in general)—does not seem to represent resource scarcity effectively, as intermittent fasting is a voluntary practice for many. Additionally, the inclusion of variables such as ethnicity, BMI, income, and sleep duration in Table 1 lacks clear relevance to the study's main focus. For future research, I suggest the authors concentrate on a specific topic and position their work within a concrete research stream.

6. A lack of methodological rigor. The article presents itself more as a conceptual paper than an empirical research-based one. This is especially evident in the Results section. The results are not reported comprehensively, making it unclear what analyses were performed, what software was used for data analysis, and other methodological details. As a reader, I would be unable to replicate the findings and procedures, rendering the current findings non-replicable and limiting the methodological rigor of this research. For example, how were sleep hours recorded? How was the sample size justified? If the fasting manipulation failed, as evident from the data, and there were no differences across the "fasting" conditions, why weren’t these two groups merged to increase the power to detect significant differences between fasting and non-fasting conditions (if any)? There are many more methodological issues in this manuscript, and its potential resubmission should be consulted with a statistician.

Reviewer #2: Summary

Landini et al. studied the differences of satiety in a relatively large cohort of participants undergoing an acute fasting intervention. The study adds important value to the field of modern fasting research, finding that the acute state of fasting (avg.8-10 h post last meal) does not affect the cognitive ability of the participants in a Raven's Progressive Matrices test. While interesting to the intersection of nutrition and cognition, I have a few comments that should be addressed before considering the manuscript for publication.

Comments to the authors:

- The terminology of long/short fast is difficult. As the authors stated, the groups did not differ in their actual fasting length. Moreover, long-term fasting is usually used as a term to describe very low calorie diets > 48 h. Hence, I strongly recommend re-working the terminology of the interventions throughout all of the manuscript.

- I like the idea of giving a protein shake to the control group. Could you provide the calories and composition of the drink?

- Most fasting-type interventions are studied in the context of metabolically unhealthy people or cohorts of advanced age. Do the authors think that the outcome would differ in older people?

- Since the effects of fasting likely intersect with circadian rhythms underlying metabolic and cognitive alterations throughout the day, it would be informative to provide the time of day when the tests were performed.

- Please add a limitation chapter discussing, e.g., the lack of difference between the groups, the short intervention time, differences to long-term interventions, the lack of cognitive disability in the young cohort, only one test for cognitive function performed, etc.

- Since there was no significant difference in fasting length between the intervention groups, the authors could combine the data and test it against the control group.

- Table 2 legend states: “a Mean comparison between control and treated, b Kruskal-Wallis test for balance across fasting treatments.” However, no p values or mean comparison are given in the manuscript.

- It would be important to state the differences between acute fasting and the outcome of weeks-long intermittent fasting/caloric restriction on cognitive ability more clearly early in the manuscript.

6. PLOS authors have the option to publish the peer review history of their article (what does this mean?). If published, this will include your full peer review and any attached files.

Reviewer #1: No

Reviewer #2: No

---

## [Author Response · Author response to Decision Letter 0]

11 Oct 2024

Below I have copied the text of our response letter, which you can also find in our attachments. Thank you again for your feedback!

To Dr. Bhattacharya and the referees at PLOS ONE,

Thank you for your time and care in reviewing our manuscript. We believe your suggestions have allowed us to greatly improve the quality of our work. In our revision process we have considered each of the comments made by the referees. We list these comments and detail our responses in the space below. As required by the editor, we are submitting our revisions using PLOS ONE’s style requirement and the LATEX template provided. We have added our full ethics statement in the Methods section of our manuscript and revised all citations and supporting information. In addition, we removed the citation of the unpublished paper. Finally, we have made our protocol available in a public GitHub repository which stores the data file from our experiment and analysis. 

Responses to Referee #1:

1. Manipulation failure. There was significant overlap in the fasting durations reported by participants in the short and long fasting groups, which may have diluted the differences between treatments. This overlap could be a result of participants' non-compliance with fasting instructions or variability in individual metabolism and meal timing. As such, the manipulation has failed. Relatedly, I did not see a solid argumentation behind why these specific desired fasting durations (3 vs 12 hours) were selected. Is there anything specific that happens in human physiology after having fasted for 3 hours relative to 12 hours? Even more importantly, is abstaining from eating for 3 hours indeed a “fasting”? 

We have added additional justification for the fasting times chosen as treatments in our ‘Experimental Design’ section as follows: 

The 12-hour fast reflects recent research suggesting that at about 12 hours of fasting there is a metabolic shift from lipid/cholesterol synthesis and fat storage to mobilization of fat through fatty acid oxidization and fatty-acid derived keytones (Anton, et al., 2016). This metabolic shift could induce fatigue or other factors which might impact performance. Even before any metabolic change has time to take effect, it is possible that hunger may affect decision making or cognitive performance because it is associated with activation of brain areas that are disproportionately activated when immediate rewards are available (Ashton 2015) . Participants in the short fasting condition were asked to refrain from eating for at least three hours prior to their session with the goal of mimicking the state of hunger that individuals often experience in between meals, similar to the judges studied in Danziger, et al., (2011).

As manipulation checks, our participants self-reported number of fasting hours and level of hunger in the post experiment survey. While the fasting treatments failed to create a significant difference in fasting hours or hunger between the two levels of fasting, the experiment did succeed in manipulating the hunger levels of participants as compared to the control group (difference significant at the p<.01 significance level), and participants in the two treated groups fasted for 9.2 hours (s.e. 3.83 hours). Following the suggestions of both referees, we have combined the treatment groups in our main analysis.

2. Limited scope of cognitive tasks. The study focused solely on the Raven's Progressive Matrices test, which is a relatively narrow measure of the cognitive ability. A stronger justification behind the sole focus on this specific task is needed to build an argument that fasting might (or might not) affect cognitive capabilities. What about other aspects of cognitive ability such as, for instance, discount rate or impulsiveness? In sum, This narrow focus might not capture other aspects of cognitive function, such as memory, attention, or executive function, which could also be affected by fasting.

We recognize this as a limitation of our work and have now added a ‘Limitations’ section following the results. In this section, we comment on the narrow scope of the Raven’s test:

`While the Raven's progressive matrices test has been widely used in previous studies of cognitive ability, it is limited in its scope and may be a relatively narrow measure of cognitive ability. The test measures fluid intelligence, which is a non-verbal estimate of abstract reasoning, cognitive thinking, problem-solving, and pattern recognition skills. Other studies (see: Levy, et al, 2013; Kuhn, et al., 2014; deRidder, et al., 2014, Ashton, 2015) have highlighted other aspects of cognitive ability such as discount rate or impulsiveness which may be salient in overall cognition. Other factors such as memory, attention, or executive function, could also be affected by fasting. This study is limited by the number of tasks we could reasonably ask our respondents to complete during the experimental session, however, future studies on the impact of fasting on cognitive ability should consider aspects of the relationship other than logic or fluid intelligence.’

3. Unclear role of IAT. This is related to point above. When reading the manuscript, I thought that the implicit association test (IAT) will be yet another outcome variable, potentially broadening the scope of the investigation. However, the results section does not focus on IAT at all.

Our original hypothesis was that implicit association could be an important mediator of the relationship between short-term fasting and cognitive ability. In particular, we hypothesized that individuals with a strong mental association between unhealthy and tasty food might be more impacted by hunger created by fasting. We have added the following statement in the appendix which includes additional detail about the IAT:

“We hypothesized that individuals' implicit associations between unhealthy and tasty food might act as a mediator of the relationship between short-term fasting and cognitive ability. However, since we do not find any significant main effect to mediate, implicit association is instead retained as a control variable in our analysis.”

4. Claims not supported by the findings. For example, the authors claim in the abstract that their results contribute to the literature on the health implications of short-term fasting, despite finding null treatment effects on cognitive ability. It is unclear how the current findings contribute to this research area, especially considering the fasting manipulation did not produce significant results. I found many other unsupported claims in different parts of the manuscript. 

Thanks for raising this issue. We have revised the manuscript to make only the claim that our results provide evidence that short term fasting does not directly inhibit cognition. 

5. The lack of the overarching research goal. The manuscript lacks a clear research goal and fails to support several claims. The role of the Implicit Association Test (IAT) is unjustified, and the operationalization of cognitive ability is limited. The introduction does not clearly position the research within a specific field of study. While resource scarcity is mentioned, the predictor variable—fasting differing in duration (albeit being short in general)—does not seem to represent resource scarcity effectively, as intermittent fasting is a voluntary practice for many. Additionally, the inclusion of variables such as ethnicity, BMI, income, and sleep duration in Table 1 lacks clear relevance to the study's main focus. For future research, I suggest the authors concentrate on a specific topic and position their work within a concrete research stream.

As noted in point 3, our original hypotheses were that these factors might act as mediators of the relationship between short-term fasting and cognitive ability. There are very few studies which test for heterogeneous treatment effects across participants. Had we found a statistically significant relationship between short-term fasting and Raven’s test score, we would have tested whether IAT or the interaction between IAT and various sociodemographic factors was important in mediating this relationship. In the end we were only able to use these variables as controls. 

The resource scarcity literature we cite focuses on `hot states’. Our hypothesis was that hunger resulting from short term fasting could produce a hot state which altered decision making, in this case altering performance on the cognitive ability test. The treatment did succeed in manipulating hunger- those in the treatment groups self-reported being hungrier than those in the control. Unfortunately, since the relationship between short-term fasting and Raven’s test score was not significant, we were not able to comment on whether individuals from certain sociodemographic backgrounds, or with especially strong unhealthy-tasty implicit association were more likely to have their decision making impacted by fasting. 

6. A lack of methodological rigor. The article presents itself more as a conceptual paper than an empirical research-based one. This is especially evident in the Results section. The results are not reported comprehensively, making it unclear what analyses were performed, what software was used for data analysis, and other methodological details. As a reader, I would be unable to replicate the findings and procedures, rendering the current findings non-replicable and limiting the methodological rigor of this research. For example, how were sleep hours recorded? How was the sample size justified? If the fasting manipulation failed, as evident from the data, and there were no differences across the "fasting" conditions, why weren’t these two groups merged to increase the power to detect significant differences between fasting and non-fasting conditions (if any)? There are many more methodological issues in this manuscript, and its potential resubmission should be consulted with a statistician.

We have revised our methods section to make the steps to replicate clearer. All covariates from the socio-demographic survey are self-reported, other than height and weight. This is now noted in the methods section. In the Appendix, we include a link to the complete formatted dataset which can be used for replication. 

To ensure that our study had sufficient power to detect meaningful differences in the number of correct responses on the Raven’s Progressive Matrices test, we conducted a power analysis based on a Poisson distribution with confidence level $\\alpha$ = 0.05. Using the control and treated means as the null and alternative hypotheses with a sample size N= 244, the results indicated that the study had an estimated power of 1-$\\beta$ = 99.57\\% to detect this difference. Thus, the study is well-powered to identify differences in cognitive performance as measured by the Raven’s test.

As both referees suggest merging the two treatment groups since there are not significant differences in fasting hours between them, we have done so in our main analysis and retained the disaggregated model in the appendix. 

We would like to provide some clarity on the choice of model used. The outcome variable, Ravens score, is a percentage calculated as % correct answers/24 possible. Because there are 24 discrete possible scores, we do not have a normal distribution of errors as is required for linear estimators to be efficient and unbiased. Logarithmic estimators do not perform well when all observations are positive, while the beta distribution is not appropriate because the score variable includes values of 1. Our original draft used a poisson estimator. This assumes mean equal to variance, which turns out not to be true. The data is underdistributed with mean larger than variance. For this reason, on our final submission we have revised to a quasi-poisson estimator: a generalized method of moments estimator, with the family poisson and a log link function to account for the only positive observations. The effect of this change is that the coefficient estimates are unchanged, but our standard errors in this draft should be more efficient as we have relaxed the mean equal to variance assumption required by poisson. 

Our data analysis section now contains a description of all steps used for data analysis:

First, we calculate a test score by dividing the subject's number of correct answers by a total 24 possible questions. Questions which were not answered are counted as incorrect responses. To prepare our data for analysis, we combined the results of the Raven's test with IAT test D-Score, recorded height and weight, and self-reported sociodemographic survey results. We calculate BMI using the standard formula: $\\frac{weight}{height^2}*703$. The complete merged data including all self-reported covariates is available upon request to the corresponding author. No identifying information about respondents is included in the final dataset. 

While the Raven's score is calculated as a percent, it is not a continuous variable, rather it reflects counts of correct answers. As a result, the errors are not normally distributed and a linear model is not the best fit. Instead we estimated a Quasi-Poisson regression model with a log link function to model the expected number of correct answers on the Raven’s Progressive Matrices test.

Referee #2

1. The terminology of long/short fast is difficult. As the authors stated, the groups did not differ in their actual fasting length. Moreover, long-term fasting is usually used as a term to describe very low calorie diets > 48 h. Hence, I strongly recommend re-working the terminology of the interventions throughout all of the manuscript.

We appreciate this feedback and instead of using the short/long terminology we switch in all cases to the specific fasting times (3-hr and 12-hr). The rationale for choosing these fasting durations are justified in the Experimental Design section (see response to Editor #1 point #1 above). 

I like the idea of giving a protein shake to the control group. Could you provide the calories and composition of the drink?

We have added the following description to our Experimental Design section: 

“The Special K brand breakfast shake contained 30g Protein, 190 Calories, 5g Total Fat, 24g Total Carbohydrate and 18g Total Sugars.”

3. Most fasting-type interventions are studied in the context of metabolically unhealthy people or cohorts of advanced age. Do the authors think that the outcome would differ in older people?

This is an interesting thought, and we add the following discussion to our Limitations section:

“Our sociodemographic survey included respondent’s age. While most participants were college students, the sample also included general population participants. There may be some concern that college students are more accustomed to meal skipping throughout the day and therefore less sensitive to the fasting times in this study. We do find a negative relationship between the age of our respondents and their score on the Raven's test indicating that it is possible that the cognitive response of older respondents may be more severely impacted by fasting. It is also possible that cognitive ability itself is correlated with age. Unfortunately, we do not have the power to test effects by age. Future studies should continue to study this type of heterogeneous treatment effect across respondents.”

4. Since the effects of fasting likely intersect with circadian rhythms underlying metabolic and cognitive alterations throughout the day, it would be informative to provide the time of day when the tests were performed.

We include the following discussion of this point in our Appendix: “Most experimental sessions were conducted in the morning. As a result, assignment into the two fasting treatments did not create a significant difference in the average fasting hours between them. Many respondents in the 3-hour fast did not wake up and eat at least 3-hours before the session, resulting in fast times closer to the 12-hour overnight fasting group. In our main analysis the treated groups are combined into one category for analysis. As a robustness check, we disaggregate the treatment groups in columns (1) and (2) of Table 4. We provide two additional specifications for robustness (

---

## [Editor Report · Decision Letter 1]

15 Oct 2024

Food for Thought: The Impact of Short Term Fasting on Cognitive Ability

PONE-D-24-28077R1

Dear Dr. Landini,

We’re pleased to inform you that your manuscript has been judged scientifically suitable for publication and will be formally accepted for publication once it meets all outstanding technical requirements.

Kind regards,

Joydeep Bhattacharya

Academic Editor

PLOS ONE

---

## [Editor Report · Acceptance letter]

23 Oct 2024

PONE-D-24-28077R1 

PLOS ONE

Dear Dr. Landini, 

I'm pleased to inform you that your manuscript has been deemed suitable for publication in PLOS ONE. Congratulations! Your manuscript is now being handed over to our production team.

Kind regards, 

on behalf of

Dr. Joydeep Bhattacharya 

Academic Editor

PLOS ONE